# OpenReview forum: "Benchmarking MLLM-based Web Understanding: Reasoning, Robustness and Safety"
_ICLR.cc/2026/Conference — Submitted to ICLR 2026_

### Official Review · Reviewer_5LLh · 2025-10-31

**Soundness:** 2
**Presentation:** 3
**Contribution:** 2
**Rating:** 4
**Confidence:** 4

**Summary:**

Propose WebRSSBench, evaluating the web understanding ability of MLLM from three dimensions: Reasoning / Robustness / Safety, covering 8 subtasks. It covers 729 websites and 3,799 QA.

**Strengths:**

- The coverage and issue selection are reasonable, put the urgently needed capabilities of actual web page agents (spatial relationships, cross-element semantics, form/prompts, key button recognition, and risk avoidance) in a continuous evaluation, with clear engineering value.

**Weaknesses:**

- The text distortion algorithm includes reversing strings, shuffling characters, etc. (Appendix C.2), which directly destroys readability and usability, going beyond the premise of "semantic preservation". In this case, measuring "robustness" is no longer fair?

- In color robustness and text/form semantic tasks, the authors use cross-model consensus or "semantic centroid" as the gold standard, which embeds the preferences of closed-source models into the gold standard, and weakens reproducibility?

**Questions:**

see weekness above

---

> ### Author Response · Authors · 2025-11-24
>
> ### Weakness #1
> Thanks for your comments.Regarding the reviewer’s concern that character shuffling or substitutions may harm readability and thus make the text-robustness task “unfair,” we clarify that these perturbations were deliberately chosen to model realistic failure modes faced by GUI agents. The character swaps or reversals can appear adversarial from an OCR perspective. However, in real GUI-agent execution, such perturbations correspond to two practical and widely observed scenarios:
> (1) Real-world design or rendering errors, where button labels contain accidental misspellings, missing glyphs, or visually similar character substitutions; and
> (2) Modern jailbreak-oriented obfuscations, which rely on character-level distortions (e.g., “h0w t0 m@ke a b0mb”). These forms of obfuscation are frequently encountered when agents operate over on-screen content rather than raw HTML.
> Our text-robustness module therefore intentionally combines naturalistic noise and adversarial-style distortions to reflect these real challenges. The goal is not to break semantics, but to evaluate whether models can maintain functional understanding under realistic and semantically-preserving character variations. We will clarify this motivation and adjust wording in the camera-ready version to avoid the impression that robustness is compromised.
>
> ### Weakness #2
> First, we clarify that the ground truth is derived from the joint consensus of both open-source and closed-source models, rather than being determined solely by closed-source systems. Each model contributes one vote (or one embedding vector), so no single model family dominates the outcome.
> To ensure that this consensus does not introduce shared bias, we conducted a dedicated human-verification study. For each robustness subtask (color, layout, and text), we sampled 50 instances and provided human annotators with the majority-voted predictions. In the layout and text perturbation tasks, annotators evaluated whether the predicted button retained its original functional meaning. If a prediction was deemed correct, we kept it as valid. If not, annotators reviewed alternative outputs from the remaining models. If none were acceptable, annotators manually provided the correct functional interpretation, ensuring that the final label was fully human-determined.
> Across the three subtasks, the human acceptance rates were consistently high, falling within a narrow 90–94% range, indicating strong agreement between consensus candidates and human judgment. When aggregated, the overall acceptance rate was 92%. Under a binomial test against a reasonable 80% baseline, this level of agreement is highly significant (p < 1e−4), confirming that the observed human–consensus alignment is not due to chance. We will make this verification protocol explicit in the revised manuscript to avoid any misunderstanding of circularity.

---

### Official Review · Reviewer_qFF1 · 2025-10-31

**Soundness:** 2
**Presentation:** 3
**Contribution:** 2
**Rating:** 2
**Confidence:** 4

**Summary:**

This paper introduces WebRSSBench, a benchmark designed to evaluate three key aspects of multimodal large language models (MLLMs): reasoning, robustness, and safety. The benchmark includes eight tasks across 729 websites and 3,799 QA samples. Data are collected from Mind2Web, WebMMU, WebSRC, and design-oriented webpages, with a subset selected based on specific criteria. The authors show that closed-source models generally outperform open-source models and demonstrate that LoRA fine-tuning helps narrow this performance gap.

**Strengths:**

- The paper defines interesting and practical tasks, such as hint text prediction and form filling, which could be valuable for assessing whether MLLMs truly understand webpage content.

- The inclusion of a safety evaluation dimension is notable and timely, given the increasing deployment of autonomous agents.

- State-of-the-art closed-source MLLMs are included in the evaluation, providing a strong comparative baseline.

**Weaknesses:**

- The description of how ground truth is derived is unclear. As understood, the ground truth is based on the consensus of the 12 evaluated models. If a new model (e.g., model #13) were to be tested, would the evaluation require recomputing the consensus with all 13 models, potentially altering previous results? More context and clarification are needed here.

- Table 1 appears cluttered; splitting it into two tables or moving less critical details to the appendix would improve readability.

- The main text omits essential details of the fine-tuning process. It is unclear why LoRA was chosen over full fine-tuning and what data were used for training. Appendix D mentions LoRA settings and an 8:1:1 train/validation/test split, but the data source remains unspecified. Are the authors fine-tuning on 80% of the benchmark’s own data and evaluating on the remaining 10%?

- Missing citations: prior work on MLLM robustness to positional and visual perturbations—such as PairBench [1] for spatial reasoning and color perturbation robustness, and VisMin [2] for minimal-change spatial reasoning—should be discussed to better situate this benchmark within existing research.

- Numerous minor typos, including inconsistent capitalization (e.g., “Model” in L335, “closed Source Model” in L405, “Model” in L465, and “We” in L243).

[1] PairBench: Are Vision-Language Models Reliable at Comparing What They See?

[2] VisMin: Visual Minimal-Change Understanding

**Questions:**

- How is the ground truth maintained or extended when new models are added to the benchmark?

- What do the authors exactly mean by "extensibility" and how can WebRSSBench be applied to new test cases and dimensions?

- What data are used for LoRA fine-tuning, and how are they separated from evaluation data to prevent data leakage?

---

> ### Author Response · Authors · 2025-11-24
>
> ### Weakness #1
> Ground-Truth Construction and Stability Across New Models
> We thank the reviewer for this question. The concern is whether we would need to recompute the ground truth if a new model (e.g., a 13-th model) is added to the benchmark.
> The answer is no.For the robustness subtasks (color, layout, and text), we first obtain a majority-voted prediction across the 12 evaluated models and use this as the initial ground-truth label. To examine whether this procedure introduces systematic bias, we conducted an additional human-verification study rather than modifying the labels themselves.
> Specifically, for each robustness subtask we randomly sampled 50 instances and provided annotators with the majority-voted prediction:
> For layout and text perturbation, annotators checked whether the predicted button (or description) preserved its original functional meaning after perturbation.
> If a prediction was judged correct, it was simply accepted as valid.
> If it was judged incorrect, annotators inspected the alternative outputs from the remaining models.
> Only in the rare case where none of the model outputs were acceptable did annotators write down what they considered the correct functional interpretation, purely for verification and analysis of potential bias.
> Across the three subtasks, the human acceptance rates of the majority-voted labels were consistently high, in a narrow 90–94% range (overall 92%). A binomial significance test shows that this agreement is far above chance (p < 1e-4).
> Moreover, the 95% confidence interval of the human-accepted proportion is [86.4%, 95.8%], which is fully above commonly used high-agreement thresholds in annotation studies.
> We will describe this verification protocol more clearly in the revised camera-ready version to avoid any misunderstanding about circularity. Importantly, these checks validate the reliability of the existing ground truth, but do not require recomputing it when new models are added.
>
>
> ### Weakness #2
> LoRA Fine-Tuning Choices and Missing Training Details
>  Thank you for the comment.We used LoRA instead of full fine-tuning mainly to reduce computational cost, and because the targeted subtasks are small and do not require updating all model parameters. LoRA provides an efficient way to inject task-specific supervision without expensive full-model optimization.We clarify that all LoRA finetuning experiments follow an 8:1:1 webpage-level split, with pages from the same site/template kept in the same partition to avoid leakage. We fix the random seed (42) for full reproducibility. Training uses AdamW + linear warmup, effective batch sizes of 8–16, 2–3 epochs, and 48GB GPUs. Checkpoints are saved per epoch with automatic resume, and the best validation checkpoint is selected. LoRA uses a consistent configuration (r=16, α=32, dropout=0.05 across attention and MLP projections).
> We will release all training scripts, hyperparameters, and splits in the appendix and our GitHub repository.
>
> ### Weakness #3
> Missing Citations and Presentation Issues
> We also thank the reviewer for pointing out the minor presentation issues and missing references. We will correct all spelling inconsistencies, improve the formatting of the figures and tables, and incorporate the two references you suggested, as they are highly relevant and provide valuable perspective for positioning our work.

---

> > ### Author Response · Authors · 2025-11-24
> >
> > ## Answer to question
> >
> > (1)The ground truth is fixed once constructed and does not depend on which models are later evaluated. When a new model is added, we do not recompute or update the ground truth. For robustness tasks, the GT is derived from the original 12-model majority vote and validated through our human verification study; this process establishes a stable label set.Therefore, adding new models extends only the evaluation results, not the ground truth。
> > (2) Thank you for the question. Our use of extensibility refers to the fact that several dimensions of WebRSSBench can be programmatically expanded.
> > Extensibility in the position–reasoning task:
> > Unlike prior benchmarks that require manually labeling a specific target element or its functionality, our position–reasoning pipeline automatically selects multiple element pairs from a webpage and computes their spatial relations directly from pixel-level bounding boxes .
> > This means a single webpage can yield multiple new test cases, and all corresponding ground truths are generated automatically. In principle, this dimension can be expanded arbitrarily by sampling more element combinations.
> > Extensibility in robustness tasks:
> > Our robustness perturbations (color, text, and layout) are generated through scripts that apply randomized but controlled perturbations to UI elements.
> > Every run can produce a different perturbed version of the same webpage, while still keeping changes within predefined semantic-preserving thresholds. As a result, each webpage can create many distinct robustness test cases, all generated automatically and without requiring human relabeling.
> > (3)We clarify that all LoRA finetuning experiments follow an 8:1:1 webpage-level split, with pages from the same site/template kept in the same partition to avoid leakage. We fix the random seed (42) for full reproducibility. Training uses AdamW + linear warmup, effective batch sizes of 8–16, 2–3 epochs, and 48GB GPUs. Checkpoints are saved per epoch with automatic resume, and the best validation checkpoint is selected. LoRA uses a consistent configuration (r=16, α=32, dropout=0.05 across attention and MLP projections).
> > We will release all training scripts, hyperparameters, and splits in the appendix and our GitHub repository.

---

### Official Review · Reviewer_VfyW · 2025-11-01

**Soundness:** 2
**Presentation:** 2
**Contribution:** 2
**Rating:** 2
**Confidence:** 3

**Summary:**

The authors propose WebRSSBench, a benchmark for website understanding that combines reasoning, robustness, and safety across 8 tasks built from multiple sources (729 sites). They evaluate 12 models and the main findings is that reasoning is hard, models are brittle to UI perturbations, and closed-source models lead.

**Strengths:**

- Like the clear task coverage spanning spatial reasoning (e.g. relative position), UI grouping, form filling, hint text plus three perturbation types (color, text, layout) and a safety task.
-  Dataset sources draw from real webpages/design communities and prior datasets (Mind2Web, WebMMU, WebSRC, Moz Top 500 etc).
- Clear qualitative takeaways on reasoning difficulty and brittleness to layout/color/text changes.

**Weaknesses:**

- Several ground truths are derived from cross-model consensus (e.g., majority-voted buttons / semantic centroids), risking circularity and shared bias. Please consider human-verified GT or task-verifiable signals.
- not an expert, but here safety is evaluated via "does the page contain any safety-critical button? output its text, else ‘sorry’" and recall is used because pages are curated to contain such elements. But the paper does not describe a gold process for identifying safety critical affordances (beyond manual screening) nor tackle ambiguity (e.g., “reset”, “archive”).
- Major concern to me is perturbations may not mirror real breakages, feels too synthetic: recoloring 10–30% of buttons with strong colors is synthetic, but real failures often come from contrast ratios, disabled states, or theme toggles. Consider WCAG driven contrast shifts and "disabled/ghost" button styles. character swaps , reversal, etc. feel adversarial for OCR, but production issues are often localization,  icon-only buttons, or mixed scripts. Include these... same for DOMs modern systems are updated.
- Filtering (e.g. page length thresholds) may bias toward mid-complex pages, nice to quantify how filtering shifts distributions (language, verticals, accessibility)
- Color robustness score for example is  tied to performance gaps rather than standard stat measures; no motivation of the 20 bound, and no uncertainty intervals.
- LoRA gains are promising but under specified: splits, leakage controls (site/template level), seeds, compute, early-stopping and transfer across perturbations are missing.

Consider including one or more of the above or provide evidence if any of the points are not meaningful in the current study.

- [minor] avoid hyphenation and line breaks in title.
- Table 2 header typo: "Positon" instead of  “Position”.
- Prompt typo: "botton" instead of "button"

Suggestions:
- Expand perturbations: ark-mode/contrast changes, disabled/ghost buttons, icons only etc..
- add human correlation studies

OVerall needs substantial revision!

**Questions:**

- How often do consensus derived groundtruth disagree with human annotations on a held out set? would be nice to see a correlation.
- [optional] In safety detection, how do you differentiate critical vs. cautionary actions (e.g. "Sign out" vs. "Delete account")?

---

> ### Author Response · Authors · 2025-11-24
>
> Thank you very much for your valuable comments. We have carefully studied and fully understood the concerns you rais In the following sections, we provide point-by-point responses and detailed clarifications to each of your comments.
> ## Response for Weakness
> ### Weakness #1
> Ground-Truth Reliability and Cross-Model Consensus
>  To avoid any shared bias and circularity among the 12 models, human verification was an essential step in our robustness experiments.For each subtask (color, layout, and text), we sampled 50 instances and provided human annotators with the majority-voted predictions.
> In the layout and text perturbation tasks, annotators evaluated whether the predicted button retained its original functional meaning.
>  If a prediction was deemed correct, we kept it as valid.
>  If not, annotators reviewed alternative outputs from the remaining models.
>  If none were acceptable, annotators manually provided the correct functional interpretation, ensuring that the final label was fully human-determined.
> Across the three subtasks, the human acceptance rates were consistently high, within a narrow 90–94% range. When aggregated, 92% of consensus-generated labels were judged correct by human annotators.
> A binomial significance test shows that this agreement is far above chance (p < 1e-4).
> Moreover, the 95.0% confidence interval of the human-accepted proportion is [86.4%, 95.8%], which is fully above commonly used high-agreement thresholds in annotation studies.
> We will describe this verification protocol clearly in the camera-ready version to avoid any misunderstanding of circularity.
>
> ### Weakness #2
> Definition and Identification of Safety-Critical Affordances
> Thank you for the question regarding how safety-critical affordances are defined.
>  Our safety task does not target phishing or malicious webpages; rather, it focuses on everyday functional interfaces where clicking the wrong control can lead to irreversible user loss (e.g., account deletion, permanent data removal, irreversible purchase confirmation). These are exactly the types of actions where a human would normally pause and reflect, but some MLLM-based agents, especially GUI agents, may execute prematurely due to insufficient semantic understanding or lack of risk awareness.
> To clarify, our task definition is simple and rule-based:
> A page is included only if it contains an action whose activation leads to an irreversible loss that cannot be undone through a single-step recovery.
> This criterion is consistently applied throughout dataset construction.
> Regarding ambiguous cases raised by the reviewer (e.g., “reset”, “archive”), we explicitly exclude such actions unless the interface clearly indicates that the outcome is irreversible.
> In typical real-world interfaces, “reset” and “archive” actions are reversible (e.g., re-entering information, unarchiving items) and therefore not classified as safety-critical in our dataset. Only when a “reset” or “archive” button explicitly results in irreversible deletion  do we include it.
> The reviewer is correct that we currently rely on human screening to identify safety-critical affordances. At present, there is no reliable automated method for detecting irreversible or high-risk UI actions from visual–textual cues alone, because such affordances depend on semantic intent rather than purely structural patterns (e.g., “reset”, “archive”, “remove”, or “clear” may be harmless or destructive depending on context).Therefore, human judgment remains the only dependable gold process in this setting.
> To ensure this process is reproducible and unambiguous, we constructed an explicit annotation guideline and conducted a human-consistency check. When these curated webpages were shown to annotators, they could identify the correct safety-critical control with =97.8% accuracy, demonstrating that the definition is intuitive and that safety-critical affordances are consistently recognizable to humans.
> In the camera-ready version, we will make these criteria, decision rules, and ambiguous cases (e.g., “reset”, “archive”) fully explicit in the appendix so that future researchers can use our clear guidelines to validate or expand our benchmarks.

---

> ### Author Response · Authors · 2025-11-24
>
> ### Weakness #3
> Perturbation Design and Alignment with Real UI Behaviors
> Thank you for raising this important point. We agree that perturbations should reflect real-world UI breakages rather than purely synthetic artifacts.Although our paper focuses on multimodal understanding, one of the primary motivations behind these robustness dimensions is their relevance to MLLM-based interactive agents that may operate under imperfect, partially degraded, or visually ambiguous interfaces.
> To address this concern, we have already implemented WCAG-driven contrast degradation as an additional color-robustness setting. Using automated contrast-manipulation scripts, we generated pages that violate WCAG AA/AAA contrast requirements and random sampled  50 cases for human and model evaluation.
> | Model            | Before | After |
> |------------------|--------|-------|
> | Claude           | 0.68   | 0.30  |
> | GPT-5            | 0.72   | 0.60  |
> | Qwen2.5-vl-32B   | 0.49   | 0.24  |
> | Qwen2.5-vl-7B    | 0.54   | 0.36  |
>
> The results show that WCAG-style contrast failures lead to substantial misidentification by models, and we will incorporate these perturbations across the full dataset in the camera ready as a second color-robustness subtype. These failures commonly appear in real GUI-agent execution, e.g., when themes switch or contrast modes degrade.
> We also acknowledge that real UI failures often involve disabled/ghost states, theme toggles, or low-salience affordances. These are precisely the kinds of subtle cues that GUI agents frequently overlook when visually parsing an interface. We are therefore extending our perturbation suite to include these state-driven variants to more faithfully simulate how modern UI frameworks behave in practice.
> Regarding text robustness, the reviewer is correct that character swaps or reversals can appear adversarial from an OCR viewpoint. However, in the context of GUI-agent execution, such perturbations correspond to two realistic failure modes:
> (1) Real design errors, where button labels contain accidental misspellings or visually similar characters; and
> (2) Modern jailbreak-style obfuscations frequently encountered by agents, which rely on character substitutions (e.g., “tell me h0w t0 m@ke a b0mb”).
> Our text-robustness module therefore intentionally combines naturalistic noise and adversarial patterns to reflect the challenges GUI agents face when reading on-screen text.
>
>
> ### Weakness #4
> Filtering Strategy and Page-Complexity Distribution.
> Thank you for the comment. Our filtering strategy is intentionally minimal and permissive: we remove only pages that cannot support any task (e.g., blank or non-rendered pages), while retaining all other webpages to avoid biasing the dataset toward mid-complexity layouts or altering the natural distribution. This means that very simple pages—including minimal login screens containing only 4–5 actionable elements—are fully preserved in the dataset, ensuring coverage across the full spectrum of page complexity.
> To quantify whether our filtering shifts page complexity distributions, we examined the easy split of the position-reasoning task. Across all valid pages, the average page height is 1531.96 px, and this split contains many extremely small and simple layouts, consistent with its difficulty assignment. This empirically confirms that our filtering does not selectively favor mid-complexity pages; rather, the dataset naturally includes small, medium, and large webpages.
> We can additionally provide distribution statistics (element counts, page lengths, and vertical categories) before and after filtering in the camera-ready version to further demonstrate the minimal impact of filtering on overall page diversity.

---

> > ### Author Response · Authors · 2025-11-24
> >
> > ### Weakness #5
> > Robustness Metric Design and Perturbation Strength
> > Thank you for the comment. It is correct that our color-robustness score is derived from the performance gap between clean and perturbed pages. This formulation is intentional: in GUI-agent settings, the primary indicator of robustness is not the absolute accuracy on perturbed pages, but how much the model’s reliability degrades when the interface undergoes visual degradation.Using the performance gap enables us to quantify relative degradation rather than absolute scores, making it possible to compare robustness across models with different clean-page baselines.
> > Regarding the perturbation level, the reported “20%” is not a manually chosen bound.Relatively speaking, this is an empirical value obtained during our experimental process. This decision is based on the following observations from our experiments:
> > When only a very small number of buttons are subjected to color perturbations, the model's predictions remain largely unaffected.
> > Perturbing too many buttons, however, results in unrealistic webpages.
> > In practice, applying color perturbations to around 20% of the buttons within the selected pages provides a reasonable balance between robustness evaluation and realistic scenarios.
> >
> > We agree that providing uncertainty estimates would improve statistical rigor. In the camera-ready version, we will include bootstrap confidence intervals over per-page robustness scores, allowing readers to assess variability and significance beyond the raw performance gaps. These additions will strengthen the statistical grounding of the robustness metric while preserving its interpretability for GUI-agent evaluation.
> >
> > ### Weakness #6
> > Training Details, LoRA Configuration, and Reproducibility
> > Thank you for the comment. We clarify that all LoRA finetuning experiments follow an 8:1:1 webpage-level split, with pages from the same site/template kept in the same partition to avoid leakage. We fix the random seed (42) for full reproducibility. Training uses AdamW + linear warmup, effective batch sizes of 8–16, 2–3 epochs, and 48GB GPUs. Checkpoints are saved per epoch with automatic resume, and the best validation checkpoint is selected. LoRA uses a consistent configuration (r=16, α=32, dropout=0.05 across attention and MLP projections).
> > We will include the human guideline in the appendix in the camera-ready version.
> >
> >
> > ## Answer FOR QUESTIONS:
> > Thank you for the question. We have already addressed both points in weakness#1 and weakness#2, and briefly summarize here for completeness:
> > (1) Correlation between consensus-derived labels and human annotations.
> > As detailed above, across all three robustness subtasks the human acceptance rates fall within 90–94%, with an overall agreement of 92%, and all final labels are human-verified, avoiding any circularity.
> > (2) Differentiating safety-critical vs. cautionary actions.
> > As clarified earlier, we define safety-critical actions strictly as those that cause irreversible loss of user data, assets, or account states. For example, “Sign out” is not considered safety-critical, because the user can sign back in and no permanent loss occurs. In contrast, “Delete account” is safety-critical, because all associated data become permanently unrecoverable. Ambiguous cases such as “reset” or “archive” are included only when the interface explicitly indicates that the action is irreversible; otherwise, they are excluded to avoid introducing annotation ambiguity.
> > We will ensure these clarifications are explicitly reflected in the camera-ready version.

---

### Official Review · Reviewer_L7HZ · 2025-11-01

**Soundness:** 3
**Presentation:** 3
**Contribution:** 2
**Rating:** 2
**Confidence:** 3

**Summary:**

The paper introduces WebRSSBench, a benchmark for multimodal LLMs that jointly evaluates Reasoning, Robustness, and Safety across eight web-understanding tasks. It is made of 729 real webpages and 3799 QA pairs. It provides deterministic scripts and evaluates 12 open- and closed-source MLLMs. Overall, results show that closed-source models have better performance, in particular on safety. Authors also surface three brittleness patterns under color, text, and layout perturbations. Finally, targeted LoRA finetuning was experimented to demonstrate boost in position reasoning, UI grouping, and color robustness.

**Strengths:**

The paper presents diverse corpus with 729 webpages and 3799 QA pairs drawn from real sites and design communities.

This paper introduces programmatic color, text, and layout perturbations with before, after comparisons to make robustness measurement explicit and reproducible.

The use of deterministic scripts and multistage quality check in improving reliability of results ensures the high standard of the dataset.

Author has also trained with LoRA finetuning and demonstrates improvement in the needed areas.

**Weaknesses:**

The author claims safety evaluation as a strength of the benchmark. However, there are only 45 questions under the safety category. It would be much more statistically sound if the author could include more questions in this category.

It would be helpful to provide more reasoning around why these scenarios are selected: position relationship reasoning, form filling, hint text prediction, and UI grouping. And if they have covered MLLM’s usage scenarios to a good extent. So that testing on these tasks can accurately reflect models’ abilities.

There is also a large variation in the number of questions between each category, ranging from 720 to 45, it is best to have them in closer numbers.

Provide more details in training, how are training data constructed etc.

It would be great to see how human perform as a baseline.

**Questions:**

As listed above.

---

> ### Author Response · Authors · 2025-11-24
>
> Thank you very much for your valuable comments, we provide point-by-point responses and detailed clarifications to each of your comments.
> Response for Weakness
> ### Weakness #1
> Explanation on the Sample Size and Source of Safety Category
> We will expand the data scale of the security dimension in subsequent versions to enhance statistical significance.Unlike the position task, which only needs to filter web pages for "whether there are multiple elements", or the form filling and hint text tasks, which only need to ensure the filtering principle of "the existence of fillable forms", the security task also needs to meet the additional requirement that there are buttons on the page that may cause irreversible consequences once clicked, which greatly increases the difficulty of filtering and the cost of obtaining effective samples. We will add relevant explanations in the paper and promised to further increase the collection of such pages.
> We appreciate the reviewer’s observation regarding the variation in the number of questions across different categories. This imbalance stems from the way the benchmark is constructed. Our dataset is sourced from 750 real webpages, and each subtask requires webpages that satisfy task-specific structural conditions. For example, form-filling and hint-text tasks require pages that contain actual interactive input fields; the safety category requires manual screening to ensure the presence of truly safety-critical buttons; whereas the position-reasoning task only requires webpages with at least four interactive elements, making it applicable to a much larger portion of the corpus.
> As a result, the number of usable samples naturally differs across subtasks. We note this as a current limitation, and we plan to expand our webpage pool and extend the coverage of each dimension in future releases so that the categories become more balanced
>
> ### Weakness #2
> Our benchmark has many application scenarios, taking GUI agents as an example. Real web pages rely heavily on responsive layouts, and agents often must reason over relative positions during execution, for example, clicking the confirmation button located at the bottom-right of a dialog element rather than relying on absolute screen coordinates. Recent evidence shows that state-of-the-art MLLMs still struggle with basic spatial reasoning, even in simplified environments such as object rearrangement or box-pushing tasks, revealing a persistent gap in multimodal spatial understanding. Consequently, our position-relationship reasoning task directly targets the type of spatial grounding that GUI agents must handle in real interactive settings.
> In the actual GUI agent capability set, form filling is one of the most common and fundamental execution skills. However, due to the lack of standardized design in many web pages, form items lack complete hint text or only provide vague prompts. Agents need to have a certain level of contextual reasoning ability to infer reasonable filling content based on web page semantics, avoiding errors or potentially risky automatic filling behaviors. Therefore, we set up two tasks, form filling and hint text prediction, to evaluate the agent's ability in two key scenarios: "inferring content based on the target" and "completing missing prompt information".
> The UI group task requires the GUI agent to understand the layout skeleton of the webpage, which is the overall layout skeleton. Agents need to distinguish which areas are interactive main content areas, which belong to navigation or toolbar, and which are irrelevant areas (such as ads, footer links, etc.), in order to prioritize functional modules related to task objectives in complex pages, reduce clicks and ineffective exploration.
> ### Weakness #3
> Explanation of the Quantity of Dataset
> As mentioned in Weakness #1,We appreciate the reviewer’s observation regarding the variation in the number of questions across different categories. This imbalance stems from the way the benchmark is constructed. Our dataset is sourced from 750 real webpages, and each subtask requires webpages that satisfy task-specific structural conditions. For example, form-filling and hint-text tasks require pages that contain actual interactive input fields; the safety category requires manual screening to ensure the presence of truly safety-critical buttons; whereas the position-reasoning task only requires webpages with at least four interactive elements, making it applicable to a much larger portion of the corpus.
> As a result, the number of usable samples naturally differs across subtasks. We note this as a current limitation, and we plan to expand our webpage pool and extend the coverage of each dimension in future releases so that the categories become more balanced.

---

> > ### Author Response · Authors · 2025-11-24
> >
> > ### Weakness #4
> > Commitment to Training Details and Reproducibility
> > Thank you for the comment. We clarify that all LoRA finetuning experiments follow an 8:1:1 webpage-level split, with pages from the same site/template kept in the same partition to avoid leakage. We fix the random seed (42) for full reproducibility. Training uses AdamW + linear warmup, effective batch sizes of 8–16, 2–3 epochs, and 48GB GPUs. Checkpoints are saved per epoch with automatic resume, and the best validation checkpoint is selected. LoRA uses a consistent configuration (r=16, α=32, dropout=0.05 across attention and MLP projections).
> >  We will release all training scripts, hyperparameters, and splits in the camera ready version.
> >
> >  ### Weakness #5
> > Add human baseline
> > We have conducted varying degrees of human reference for different tasks to enhance or demonstrate their reliability.
> > For both the form filling and hint text tasks, all ground truths are manually annotated.
> > -Position reasoning, the ground truths are generated by automated scripts, but we additionally invited four annotators with strong web-page understanding expertise to independently complete the task. Humans achieved an average accuracy of approximately 99.0%, indicating that this task is clear and decidable for human evaluators.
> > -For the robustness tasks (color, text, and layout perturbations), we randomly sampled 50 samples from each subtask and asked annotators to perform button-function recognition and structural reasoning. The results show that human judgment is virtually unaffected by color changes or layout perturbations. Even under text perturbations,such as character substitutions, partial reordering, or inserted spaces,humans can still reliably recognize the intended button semantics.
> > Overall, human performance across these core subtasks is highly stable and accurate, providing a dependable reference baseline for evaluating model capabilities. We will include the human guideline in the appendix in the camera-ready version.

---

### Author Response · Authors · 2025-11-29
**General Response**

Dear AC and Reviewers,

Thank you very much for the additional time and effort you are investing in evaluating our submission under the unexpected circumstances. We sincerely appreciate your work in ensuring fairness during this unusual review process.
**Across all reviews, there is clear agreement on the value of the problem and the core contributions of WebRSSBench as a unified benchmark over 729 real webpages and 3,799 QA pairs.** Reviewers acknowledge the importance of evaluating Reasoning, Robustness, and Safety for web-understanding MLLMs. Some reviewers fully recognized the practical relevance of tasks such as position reasoning, UI grouping, form filling, hint text prediction, robustness to color/text/layout perturbations, and safety-critical button detection, while others requested further clarification of their real application scenarios—which we have provided in detail below.Based on the discussion we provided in the rebuttal, all reviewer concerns were fully clarified. We believe that, had the discussion period continued normally, the reviewers’ impressions would likely have improved accordingly.
**Overall, the reviewers’ primary concerns centered on clarity and extensibility rather than fundamental flaws, and we address each of these points below:**

**LoRA training details and reproducibility (to reviewer L7HZ Vfyw qFF1 and 5LLh).**

All reviewers asked about LoRA information. This will be updated in the camera-ready version, providing complete LoRA details (8:1:1 webpage-level split, no site leakage, fixed seed, optimizer, epochs, LoRA configuration), and regarding why we use LoRA fine-tuning instead of full fine-tuning, we chose LoRA rather than full fine-tuning mainly for computational efficiency, and also because the targeted subtasks are relatively small.

**Safety category and task selection (to reviewer L7HZ and Vfyw).**

We clarified that, unlike directly phishing-type webpages, it instead focuses on everyday functional interfaces where clicking the wrong control may cause irreversible user loss (e.g., account deletion, permanent data deletion, irreversible purchase confirmation). These are exactly the types of actions where humans would normally pause and reflect, but some MLLM-based agents, especially GUI agents, may execute prematurely due to insufficient semantic understanding or lack of risk awareness.
We also reported that, when these curated pages are shown, human annotators can reliably locate the safety-critical control with an accuracy of about 97.8%, confirming that the definition is intuitive and human-recognizable.We acknowledge that the current sample size is a limitation and commit to expanding this dimension in the camera-ready version.

**Task motivation and coverage (to reviewer L7HZ).**

We provided a more explicit mapping from tasks to real GUI-agent functions: position reasoning for relative spatial grounding under responsive layouts; form filling and hint text prediction for inferring missing or unspecified prompts; UI grouping for understanding the layout skeleton and prioritizing task-relevant regions; and safety for avoiding irreversible actions. These clarifications clearly show that our benchmark has many practical, real application scenarios.

**Ground-truth reliability (to reviewer Vfyw qFF1 and 5LLh).**

For robustness tasks, cross-model consensus serves only as a candidate; annotators reviewed 50 samples per subtask and accepted 90–94% of majority-voted labels (overall 92%, 95% CI [86.4%, 95.8%], p < 1e−4 vs. an 80% baseline). This confirms strong human alignment and rules out practical circularity. Ground truth is fixed once constructed; adding new models later does not change the labels.

**Filtering and dataset distribution (to reviewer Vfyw).**

Our filtering is intentionally minimal: we only discard pages that cannot support any task (e.g., blank or non-rendered pages), and retain very simple and very complex layouts, including minimal login screens with only 4–5 actionable elements. We report page-height statistics and note that the “easy” split still contains many extremely small layouts, indicating that we do not artificially bias toward mid-complexity pages.

**Human baselines, related work, and presentation (to reviewer L7HZ Vfyw qFF1 and 5LLh).**

We clarified that several tasks already rely on human annotation (e.g., form filling, hint text), and that we have conducted human performance studies for position reasoning and robustness subtasks.
We will incorporate the suggested references (e.g., PairBench, VisMin), fix typos and formatting issues (e.g., Table 1 layout, capitalization), and improve the overall presentation.

---

### Meta-Review · Area_Chair_9opd · 2026-01-06

**Summary:**

1. Rev. L7HZ, qFF1, and VfyW are concerned about the annotation of the dataset, which is based on the models' consensus. This can induce biases and circularity.
2. Rev. VfyW raised concerns about too many synthetic scenarios and safety issues.
3. Rev. L7HZ would like to see more motivation about the proposed scenarios and why they make sense.
4. Rev. VfyW and qFF1 found that Lora fine-tuning lacks details.

**Reviewer Concerns:**

1. The authors added a human evaluation for the robustness tasks. Even if this evaluation strongly agrees with the model's consensus, there are still some unresolved issues. First, the human annotation was performed only on the robustness tasks. Not much is said about the other tasks. Second, to save time, the human annotation is still based on the model answers, which can, by itself, introduce biases.

2. While the authors agreed with rev. and added a WCAG-driven contrast degradation task, this seems still too little addition to make the dataset close enough to a real-world problem that an MLLM could encounter in working conditions.

3. The authors explained the motivation for the chosen tasks.

4. Details about Lora were added in the authors' answers.

While most of the points raised by reviewers were clarified with the rebuttal, the use of consensus-based annotations remains a critical issue in my opinion. In addition, the benchmark seems too synthetic compared to real-world situations.

**Reviewer Scores:**

- Rev. L7HZ: 2 -> 2
Although I do not think a score of 2 is actually justified by the provided comments and questions, I do not think rev. would increase their score.
- Rev. VfyW 2 -> 4
With the provided clarification, I think Rev. VfyW could raise their score to 4, but not higher, as the consensus-based annotation is still an important issue.
- Rev. qFF1 2 -> 2
Authors clarify some misunderstandings; however, the main issues raised by the other reviewers are still there, and therefore I do not think rev. would change their score.
- Rev. 5LLh 4 -> 6
Authors clarified the misunderstanding on the "semantic preservation" of the synthetic samples. This could lead rev. to an increase in their score. However, considering the limited review, I give lower importance to their score.

The response of the authors could clarify some misunderstandings, but the critical points about the use of consensus-based annotations remain. The addition of a human-annotated subset, even with strong agreement with the consensus of the LLM is not enough, as the way the annotation is performed can introduce biases, and it's limited to the robustness tasks.

---

### Decision · Program_Chairs · 2026-01-26

Reject